# Chitosan Nanoparticles for Meloxicam Ocular Delivery: Development, In Vitro Characterization, and In Vivo Evaluation in a Rabbit Eye Model

**DOI:** 10.3390/pharmaceutics14050893

**Published:** 2022-04-20

**Authors:** Hebatallah B. Mohamed, Mohamed Ali Attia Shafie, Aml I. Mekkawy

**Affiliations:** 1Department of Pharmaceutics, Faculty of Pharmacy, South Valley University, Qena 83523, Egypt; dochoba2014@svu.edu.eg; 2Department of Pharmaceutics, Faculty of Pharmacy, Assiut University, Assiut 71515, Egypt; 3Department of Pharmaceutics and Clinical Pharmacy, Faculty of Pharmacy, Sohag University, Sohag 82524, Egypt; aml.mekkawy@pharm.sohag.edu.eg

**Keywords:** meloxicam, chitosan nanoparticles, polyethylene glycol 400, permeability study, anti-inflammatory activity

## Abstract

Eye inflammation is considered one of the most common co-morbidities associated with ocular disorders and surgeries. Conventional management of this condition with non-steroidal anti-inflammatory drugs as eye drops is associated with low corneal bioavailability and ocular irritancy. In the current study, we first investigated the capacity of different solvent systems to enhance the solubility of Meloxicam (MLX). Then, we prepared chitosan nanoparticles loaded with meloxicam (MLX-CS-NPs) through electrostatic interaction between the cationic chitosan and the anionic MLX using either 100% *v*/*v* polyethylene glycol 400 or 0.25% *w*/*v* tripolyphosphate solution as solvents based on the MLX solubility data. In further studies, MLX-CS-NPs were characterized in vitro and assessed for their ex vivo corneal and scleral permeability. The morphology, average particle size (195–597 nm), zeta potential (25–54 mV), and percent entrapment efficiencies (70–96%) of the prepared MLX-CS-NPs were evaluated. The in vitro release study of MLX from the selected MLX-CS-NPs showed a sustained drug release for 72 h with accepted flux and permeation through the cornea and sclera of rabbits. In the in vivo studies, MLX-CS-NPs eye drop dispersion showed enhanced anti-inflammatory activity and no ocular irritancy compared to MLX-eye drop solution. Our findings suggest the potential for using chitosan nanotechnology for ocular delivery of MLX with high contact time and activity.

## 1. Introduction

Ocular inflammation is one of the most prevalent diseases in ophthalmology that can affect different parts of the eye (anterior and posterior) with numerous symptoms, including eye redness, pain, edema, lowered ocular motility, and other pathological changes [1,2]. The treatment for ocular inflammation mainly includes topical steroids and non-steroidal anti-inflammatory drugs (NSAIDs) separately or in combination, inducing severe adverse effects with long-term therapy [3]. Conventional eye drops are the most suitable dosage form owing to their ease of application and patient compliance; however, topical application is accompanied by extraordinarily physiological defense mechanisms of the eye, including tear clearance and nasolacrimal drainage system. In addition, corneal and conjunctival epithelia’s intercellular tight junction complexes restrict drug entry. These protective mechanisms often limit the ocular bioavailability of conventional eye drops, as only 5% of the drug could be absorbed and reach the intraocular tissue after a single eye drop application. This could be overcome by frequent daily application (~6 times/day) that may cause patient inconvenience and drug toxicity [4]. Meloxicam (MLX) is one of the NSAIDs with selective cyclooxygenase-2 (COX-2) inhibition and potent anti-inflammatory activities [5]. However, MLX is practically insoluble in water and acidic pH with low oral bioavailability, irritancy, and significant gastrointestinal adverse effects [5,6,7]. A significant anti-inflammatory effect of MLX eye drop solution was previously reported in a rabbit model of acute ocular inflammation [8]. However, applying high MLX concentration directly on the ocular surface (highly innervated) may cause local irritation, especially with repeated exposure [9].

Indeed, nanoparticles as a delivery system are essential to sustain the ocular drug delivery, provide better bioavailability and efficacy, minimize ocular irritancy, and improve patient compliance. Nanoparticles provide an effective ocular drug delivery for both anterior and posterior eye parts with enhanced drug efficacy [10,11,12,13,14]. However, only a few studies formulated nanoparticles as eye drop dispersion for ocular drug delivery to the anterior segment of the eye (conjunctiva, cornea, iris, lens, ciliary body, and the anterior portion of the sclera) due to the ocular barriers [15,16,17,18,19]. Furthermore, only a single study formulated an MLX ophthalmic delivery system as a contact lens, including bovine serum albumin-coated MLX nanoaggregates for the treatment of post-cataract endophthalmitis [9]. 

Chitosan nanoparticles could improve the pre-ocular residence time owing to the mucoadhesive characteristics of chitosan that arises from the interaction between its positively charged amino groups and the negatively charged residues of sialic acid in the corneal and conjunctival mucosa [20]. Furthermore, it enhances the ocular drug penetration by opening the tight cell junctions in the corneal and conjunctival epithelial cell surfaces [21,22]. It is worth mentioning that chitosan, with its anti-inflammatory effect, could augment the activity of MLX in the current study [23,24]. Based on the previous data, we sought to formulate MLX loaded chitosan nanoparticles (MLX-CS-NPs) as eye drop dispersion to improve its ocular residence time, bioavailability, and efficacy and lower its irritancy.

To the best of our knowledge, this is the first study that developed MLX-CS-NPs eye drop dispersion intended for ophthalmic administration. After that, we evaluated the ex vivo permeation of MLX-CS-NPs through the rabbits’ cornea and sclera. Finally, we assessed the in vivo efficacy of MLX-CS-NPs eye drop dispersion compared to MLX eye drop solution for the treatment of acute inflammation in a rabbit eye model.

## 2. Materials and Methods

### 2.1. Materials 

MLX was kindly provided by Medical Union Pharmaceuticals (MUP) Co. (Abu-Sultan, Ismailia, Egypt). Low molecular weight chitosan (purity > 90%, cps viscosity 50–300) was obtained from Bio Basic Inc. (Toronto, ON, Canada). Polyvinyl pyrrolidone (PVP, average MWt = 44,000) was obtained from BDH Chemicals Ltd. (Poole, UK). Propylene glycol (PG, 0.995 mass fraction purity) was purchased from Adwic, El-Nasr Chemical Co. (Cairo, Egypt). Capsaicin, tripolyphosphate (TPP), polyethylene glycol 400 (PEG 400), hydroxypropyl beta-cyclodextrin (HPβ-CD), and Pluronic F-127^®^ (PF-127) were obtained from Sigma-Aldrich (St. Louis, MO, USA). All other chemicals and solvents were of analytical reagent grade.

### 2.2. Solubility Study of MLX

We determined the solubility of MLX in different systems by adding an excess amount of MLX (25 mg) into a stoppered glass tube containing 5 mL of each studied system as prescribed in Table 1. Tubes were then shaken at 100 stroke/min using shaking water bath (Gesellschaft Für Labortechnik, Germany) at 37 °C for 72 h. After equilibrium, samples were centrifuged (Centurion Scientific Ltd., West Sussex, UK) at 4000 rpm for 15 min, and the supernatant was filtered through a 0.45 µm membrane filter (super Acro disc). The absorbance of the filtrate was measured using a UV-Vis spectrophotometer (Shimadzu, Double-Beam Spectrophotometer 150-02, Kyoto, Japan) at 364 nm, and MLX concentration was calculated from a previously constructed calibration curve. 

### 2.3. Preparation of MLX-CS-NPs

MLX-CS-NPs were prepared using the polyelectrolyte complexation method [25,26] using different compositions, as shown in Table 2. First, chitosan solution was prepared as follows: the predetermined concentration (0.25% or 0.5% *w*/*v*) of chitosan was dissolved in either 0.5% or 1% *v*/*v* aqueous acetic acid solution, respectively, and the pH of the prepared chitosan solution was adjusted to 4.7 using sodium hydroxide (1 M). Second, MLX solution was prepared by dissolving MLX powder in either 0.25% *w*/*v* TPP aqueous solution or 100% *v*/*v* PEG 400. Then, MLX-CS-NPs were formed spontaneously after dropwise addition of MLX solution to the magnetically stirred chitosan solution (10 mL) for 30 min and then sonicated for 10 min using a probe sonicator (Q500 sonicator, Qsonica, Melville, New York, USA). Finally, MLX-CS-NPs were then washed, collected, and left to dry at room temperature (25 ± 1 °C) for further characterizations. The blank nanoparticles were prepared similarly to MLX-CS-NPs.

### 2.4. In Vitro Characterization of the Prepared MLX-CS-NPs

#### 2.4.1. Entrapment Efficiency (EE%)

The MLX entrapment efficiency was measured by the indirect method. Briefly, the freshly prepared nanoparticles were centrifuged at 14,000 rpm for 30 min. Next, the supernatant was measured using a UV-Vis spectrophotometer at 364 nm. Finally, the concentration of the MLX was calculated from a previously constructed calibration curve. The entrapment efficiency was determined from the following equation:Entrapment efficiency (%) = (Actual drug content/Theoretical drug content) × 100(1)

#### 2.4.2. Evaluation of the Average Particle Size, Zeta Potential, and Morphology

The mean size (hydrodynamic diameter (d.)), zeta potential, and polydispersity index of MLX-CS-NPs were determined in liquid suspension after dilution in double distilled water using a zeta sizer (Malvern Instrument Ltd., Worcestershire, UK). Next, the particles’ shape and surface morphology were examined using a scanning electron microscope (SEM, Hitachi S-4800, Hitachi, Tokyo, Japan). The freshly prepared nanoparticles suspension was added onto an aluminum SEM stub, dried at room temperature (25 ± 1 °C), and coated with gold using a sputter coater. Finally, nanoparticle images were captured by SEM.

#### 2.4.3. pH Determination

The pH of the freshly prepared MLX-CS-NPs dispersions was measured using a pH meter (Jenway 3510, Cole-Parmer, Stone, UK).

#### 2.4.4. Viscosity Measurements

The viscosity of the selected MLX-CS-NPs dispersion was measured at room temperature (25 ± 1 °C) using a Brookfield DV + II model LV viscometer (Brookfield Engineering Laboratories, Inc., Stoughton, MA, USA). 

#### 2.4.5. Fourier Transform Infrared Spectroscopy (FTIR) Analysis

FTIR spectra of pure powders of MLX and chitosan, PEG 400, their physical mixture, and the selected MLX-CS-NPs formulation were recorded using a Nicolet 6700 FTIR spectrometer (Thermo Fisher Scientific, Waltham, MA, USA) over a range of 4000 to 400 cm^−1^. All samples were prepared as compressed KBr disks.

#### 2.4.6. In Vitro Release Studies and Kinetic Analysis of the Release Data

The in vitro release of the selected formulation of MLX-CS-NPs was studied by dialysis method using a semi-permeable cellophane membrane (12,000–14,000 MWCO, Sigma Chemicals, St. Louis, MO, USA). One milliliter of the selected MLX-CS-NPs formulation (equivalent to 3.5 mg MLX) was placed on a presoaked cellophane membrane fixed on one side of an opened glass cylinder (2.5 cm diameter). The cylinder was immersed in a beaker containing 50 mL PBS (pH 7.4) as the release medium. The system was maintained for 72 h at 37 ± 0.5 °C in a thermostatically controlled shaking water bath at 50 rpm. Aliquots of 5 mL sample were withdrawn at a predetermined time (0.5, 1, 2, 4, 6, 12, 24, 48, and 72 h) and replaced with the same volume of fresh medium (maintained at the same temperature) for maintaining sink condition. Samples were analyzed for the drug content using a UV-Vis spectrophotometer at 364 nm against blank similarly treated. The experiment was carried out in triplicate, and the average values were calculated.

The release data of MLX from MLX-CS-NPs were fitted to three kinetic models: zero-order kinetic (Equation (2)), first-order kinetic (Equation (3)), and the Higuchi equation (Equation (4)). Regression analysis was adopted to compute the constants and correlation of data R^2^. Korsmeyer–Peppas equation (Equation (5)) was used to calculate the diffusion exponent (n). When n ≤ 0.45 represents a fickian diffusion, 0.5 ≤ n ≤ 0.8 represents a non-fickian diffusion, and 0.8 ≤ n ≤ 1 refers to a zero-order mechanism.
Q = K_0_t(2)
ln (100 − Q) = ln100 − K_1_t(3)
Q = K_H_ t^1/2^(4)
M_t_/M_∞_ = Kt^n^(5)
where Q is the released amount of drug at time t, K_0_ is the zero-order release constant, K_1_ is the first-order release constant, K_H_ is the Higuchi release rate constant, M_t_/M_∞_ is the released drug fraction at time t, and n is the exponent of release.

#### 2.4.7. Ex Vivo Ocular Permeation Study 

MLX permeation through the cornea and sclera of the rabbit eye was assessed for the selected MLX-CS-NPs formulation. Rabbits were sacrificed, and their corneas and sclerae were removed. Half a milliliter of the selected MLX-CS-NPs dispersion (equivalent to 1.75 mg MLX) was placed on a membrane (either the cornea or the sclera) fixed on one side of an opened glass cylinder (0.5 cm diameter). The cylinder was immersed in a beaker containing 25 mL PBS (pH 7.4) as the release medium. The system was shaken at 50 rpm for 48 h at 37 ± 0.5 °C. Samples were withdrawn, replaced with an equal volume of the release medium, and analyzed as previously mentioned in the in vitro release study section. Experiments were carried out in triplicates. The cumulative amount of MLX permeated through the membrane (either cornea or sclera) was determined at each time point [27]. The permeation of MLX from MLX solution in PEG 400 (3.5 mg/mL) was similarly assessed.

##### Permeation Data Analysis

Fick’s law of diffusion was used for the determination of membrane flux according to the following equation: [28]
J = dQ_t_/A.dt (6)
where J is the flux (µg/cm^2^/h), dQ_t_ is changing in the amount of MLX (µg) passing through the cornea or sclera to the receptor compartment, A is the area for diffusion (cm^2^), and dt is the time difference [29]. The permeation profiles were formed by plotting the cumulative amount of drug permeated per unit area of the cornea or sclera (µg/cm^2^) against time (h). Then, the permeability coefficient (P, cm/h) was obtained from the following equation:P = J/C_0_(7)
where C_0_ is the MLX initial concentration (µg/mL).

### 2.5. Preparation of MLX-CS-NPs Eye Drop Dispersion for the In Vivo Studies

MLX-CS-NPs (0.03% *w*/*v*) eye drop dispersion was prepared for further *in vivo* studies. Briefly, precisely weighed dried MLX-CS-NPs (equivalent to 3.5 mg MLX) were dispersed in 11.7 mL phosphate-buffered saline (PBS, pH 7.4) containing hydroxypropyl methylcellulose (HPMC, 0.2% *w*/*v*) as a viscolizer and methyl paraben (0.15% *w*/*v*) as a preservative (schematic presentation: Figure 1A). MLX-eye drop solution (0.03% *w*/*v*) was also prepared by dissolving 3.5 mg of MLX powder in one mL PEG 400, then 10.7 mL PBS (pH 7.4) containing (HPMC, 0.2% *w*/*v*) and methyl paraben (0.15% *w*/*v*) was added to the MLX solution.

### 2.6. In Vivo Studies 

Eye irritancy evaluation and anti-inflammatory activity studies of the selected MLX-CS-NPs eye drop dispersion were carried out and compared to the MLX-eye drop solution and the blank CS/PEG 400 eye drop solution. 

All animal experiments were carried out according to the ethical guidelines approved by the Institutional Animal Ethical Committee of the Faculty of Pharmacy, Assiut University (approval number: S23-21; approval date: 7 December 2021) that adheres to the Guide for the Care and Use of Laboratory Animals, 8th Edition, National Academies Press, Washington, DC, USA, and were conducted in accordance with the ARVO (The Association for Research in Vision and Ophthalmology) statement for the use of animals in Ophthalmic and Vision Research [30].

#### 2.6.1. Eye Irritancy Evaluation

The ocular tolerability, the potential ocular irritancy, and the damaging effects of MLX-CS-NPs eye drop dispersion were assessed according to the modified scoring system for ocular irritation testing (Table 3), established by the Organization for Economic Co-operation and Development (OECD) guidelines [31]. Nine male domestic healthy rabbits (weighing 1.5–2.5 kg) were divided into three groups (each group containing three rabbits):Group I:Blank CS/PEG 400 eye drop solution (drug-free).Group II:MLX-eye drop solution.Group III:MLX-CS-NPs eye drop dispersion.

Each rabbit received two drops of the treatment twice daily for three days in the right eye, while the left eye received two drops of PBS (pH 7.4) as control. The ocular conditions (animal discomfort and clinical signs in conjunctiva, cornea, and lids) were evaluated at different application times (0.08, 0.16, 0.5, 1, 6, 12, 24, 48, and 72 h). 

#### 2.6.2. Anti-Inflammatory Activity Study

The ocular anti-inflammatory activity of the selected MLX-CS-NPs eye drop dispersion was evaluated. Acute ocular inflammation of the rabbits was induced by applying two drops of 5% *w*/*v* capsaicin extract in the right eye only, every hour for three hours [31,32,33]. Nine male domestic rabbits (weighing 1.5–2.5 kg) were divided randomly into three groups (each group containing three rabbits): Group I:Blank CS/PEG 400 eye drop solution (drug-free)Group II:MLX-eye drop solution.Group III:MLX-CS-NPs eye drop dispersion.

Each rabbit received two drops of the treatment thrice daily for three days in the right eye, while the left eye received two drops of PBS (pH 7.4) as control (without induction of inflammation). 

The induced ocular inflammation was visually observed and graded before the start of treatment and 1, 2, and 3 days after treatment. The assigned score for corneal opacity and ulcerations ranged from 0 to 4, swelling and hyperemia of the iris ranged from 0 to 2, conjunctiva redness and vessel discernibility ranged from 0 to 3, while swelling and lids closed/open ranged from 0 to 4.

#### 2.6.3. Histopathological Examination

To assess the pathological alteration of the ocular tissues after the anti-inflammatory study, rabbits were sacrificed, and the right eyeballs were surgically isolated and fixed in 10% *v*/*v* formalin for histological examinations. Paraffin blocks were prepared, and 5 µm sections were cut, stained with Harris’s hematoxylin and eosin, and examined using a light microscope. The pathological alterations in the ocular tissues (cornea and sclera) were observed, and the histopathological appearance of the corneal epithelium was semi-quantitatively graded according to (Table 4) [34,35].

### 2.7. Statistical Analyses

Finally, we used GraphPad Prism for Windows 8.3.0 (GraphPad Software Inc.) for the statistical analyses. One-way analysis of variance (ANOVA) followed by a Tukey post hoc test was conducted to evaluate the differences in the average particle size and drug entrapment efficiencies between different MLX-CS-NPs formulations. In addition, the in vivo anti-inflammatory activity of MLX-CS-NPS eye dispersion, MLX-eye drop solution, and blank CS/PEG 400 eye drop solution was also statistically evaluated.

## 3. Results and Discussion

### 3.1. Solubility Study of MLX 

First, we sought to increase MLX solubility in water using different hydrophilic polymers or combinations. Polymers were preferably selected over surfactants for enhancing MLX solubility in ophthalmic preparations such as eye drops as they show no irritation, toxicity, or hemolysis. Moreover, they increase the ocular retention time of the drug solution by increasing its viscosity [36]. As demonstrated in Table 5, the apparent solubility of MLX was dramatically increased with all the studied systems compared to its previously reported water solubility (12 µg/mL) [36,37,38], especially with 100% *v*/*v* PEG 400 and 0.25% *w*/*v* TPP it was 3.8 mg/mL and 1.9 mg/mL, respectively. This tremendous increase in MLX solubility with PEG 400 could be attributed to the possible hydrophobic interaction of MLX with hydroxyl groups (OH) in the polyethylene chains of PEG 400 that lead to hydrogen bonding formation [39,40,41,42]. Furthermore, the solubility of MLX was increased with increasing PG and TPP concentrations. Similarly, MLX solubility was enhanced by increasing the concentration of HP-β-CD either alone or in combination with other polymers (5% *w*/*v* PF-127 and 1% *w*/*v* PVP), which could be attributed to its strong complexation capacity towards MLX [36]. 

### 3.2. Preparation and Characterization of MLX-CS-NPs

MLX-CS-NPs were successfully prepared using the polyelectrolyte complexation method. First, MLX was solubilized in either 0.25% *w*/*v* TPP aqueous solution or 100% *v*/*v* PEG 400 (the best solvent systems selected from the above solubility study). MLX solutions were then incorporated into chitosan solutions to prepare MLX-CS-NPs. Finally, chitosan nanoparticles were formed through the ionic interactions between the protonated amino groups of chitosan and either the tri-polyphosphoric ions of TPP [43] or MLX itself, which served as the anionic cross-linker; this process is known as drug-induced gelation [44]. 

#### 3.2.1. Morphology, Entrapment Efficiency, Average Particle Size, and Zeta Potential Measurements

The freshly prepared MLX-CS-NPs showed spherical-shaped particles with a smooth surface under SEM (Figure 1B). The entrapment efficiency of MLX was significantly increased (*p* < 0.001) in F3 and F6 (96% and 91%, respectively) compared to other formulations. Generally, the nanoparticles’ size and surface charge intended for ocular delivery are highly considered as penetration of particles into the ocular surface depends on their size, surface charge, and architecture [45]. The mean particle diameter, polydispersity index, zeta potential, EE %, and pH of MLX-CS-NPs formulations are presented in Table 6. The average particle size of the drug-loaded nanoparticles changed significantly (*p* < 0.001) between different compositions of the nanoparticles and ranged between 195 and 597 nm. MLX-CS-NPs (F3 and F6) containing PEG 400 exhibited a significant decrease (*p* < 0.001) in the particle size (195 and 242 nm, respectively) when compared to other formulations (including 0.25% *w*/*v* TPP). This is most likely attributed to the higher solubility of MLX in PEG 400 with a stronger binding affinity over TPP. Moreover, the steric hindrance of PEG 400 limits the intermolecular cross-linking between chitosan molecules, thus producing nanoparticles of smaller size [46]. All the prepared MLX-CS-NPs had positive surface charge with zeta potential values ranging from 17 to 57 mV, which is considered high enough to prevent particles aggregation [47]. F3 and F6 showed lower zeta potential value compared to other formulations, which might be due to the high entrapment amount of MLX. Based on the previous characterization, F3 was selected for further characterizations as it showed the highest EE% (96%), the smallest average particle size (195 nm) with a low polydispersity index (0.41), accepted surface charge (28 mV), and suitable viscosity (6.8 cps at 50 rpm and 30 °C).

#### 3.2.2. Fourier Transform Infrared Spectroscopy (FTIR)

FTIR spectra of MLX, PEG 400, chitosan (CS), their physical mixture (PM), and MLX-CS-NPs are shown in Figure 2. The most characteristic band in MLX is due to the secondary amine stretching (3292 cm^−1^) and is still present in the same region in the FTIR spectrum of the physical mixture but with small intensity [48,49]. This band was almost disappeared in the spectrum of MLX-CS-NPs, indicating the encapsulation of MLX within the polymeric matrix [50]. No new bands were observed in both spectra of the physical mixture and NPs, which suggests no chemical interactions between the drug and the preparation components [2].

#### 3.2.3. In Vitro Release and Kinetic Studies

Further in vitro evaluation of the selected MLX-CS-NPs (F3) was carried out. The release profile of MLX from F3 formulation was studied and compared to that from MLX solution in PEG 400 (Figure 3). The results showed that MLX solution in PEG 400 exhibited a high initial drug release in the first hour (~16%) and a complete drug release after 12 h. This fast release implies that several administrations would be necessary to treat the eye inflammation. On the contrary, MLX-CS-NPs exhibited a prolonged drug release profile. The release profile of MLX from MLX-CS-NPs showed a biphasic release pattern, an initial rapid phase followed by a slower release phase. The initial immediate release is due to the superficially adsorbed MLX molecules, while the sustained release was mainly caused by the diffused MLX through the chitosan polymer matrix [4,48,51]. 

Kinetic analysis of the release data of MLX from MLX-CS-NPs dispersion and MLX solution in PEG 400 was performed using different kinetic models, and the results are shown in Table 7. The mechanism of release of MLX was assigned to the kinetic model with the highest value of the calculated correlation coefficient (R^2^). The release data of MLX from both MLX-CS-NPs dispersion and MLX solution in PEG 400 were explained by diffusion and zero-order models, respectively. The release exponent “n” was obtained from the Korsmeyer–Peppas equation and indicated an anomalous non-fickian diffusion of the drug from MLX-CS-NPs dispersion (*n* = 0.69). Our results are in concordance with the previously reported kinetic studies of different drugs release data from chitosan nanoparticles [52,53].

#### 3.2.4. Ex Vivo Corneal and Scleral Permeability 

The permeation study of MLX through the cornea and sclera of rabbit eye from the selected MLX-CS-NPs (F3) and MLX solution in PEG 400 was carried out, and the results are shown in Table 8. MLX-CS-NPs exhibited a sustained release of the drug over the MLX solution in PEG 400, which was caused by the increased pre-corneal retention time of the drug-loaded chitosan nanoparticles (Figure 4). However, the flux and permeability of MLX from MLX-CS-NPs through cornea and sclera showed low values compared to that of the MLX solution in PEG 400 (Table 8). This could be explained by the entrapment of MLX inside the chitosan matrix of MLX-CS-NPs; thus, its flux and permeability were slow with a sustained release (72 h). On the contrary, MLX from the MLX solution in PEG 400 could permeate freely through the cornea and sclera. Notably, the amount of MLX that permeated through both cornea and sclera was highly accepted regarding the reported effective dose of MLX for treating ocular inflammation [9]. 

### 3.3. In Vivo Studies

#### 3.3.1. Eye Irritancy Assessment

The ocular irritation tendency of the blank CS/PEG 400 eye drop solution, MLX-CS-NPs eye drop dispersions and MLX-eye drop solution was evaluated. The medicated nanoparticles and the blank solution showed no changes in the cornea, iris, and conjunctiva with no eye secretions (total irritancy score = zero). However, redness and mild secretions with a bit of blinking were observed after applying MLX-eye drop solution (total irritancy score = 0.3 ± 0.2). Our results suggested that the provided chitosan nanoparticles as an ocular drug delivery system can overcome the irritant side effect of MLX chronic use. Therefore, the selected MLX-CS-NPs (F3) are considered a safe formulation for ophthalmic application and could be further subjected to the *in vivo* anti-inflammatory study.

#### 3.3.2. In Vivo Anti-Inflammatory Study 

The most affected part of the eye after 5% capsaicin-induced inflammation is the anterior chamber, which is preferably treated using an eye drop formulation. The anti-inflammatory activity of the selected MLX-CS-NPs eye drop dispersion (F3) was evaluated and compared to the blank and MLX-eye drop solutions. The results showed a significant reduction (*p* < 0.001) in the inflammation score from the first day of treatment after application of MLX-CS-NPs eye drop dispersion and complete recovery after three days, while a more extended time was recorded for full recovery in the MLX-eye drop solution treated group (Figure 5 and Table 9). Beyond the fact that nanoparticles act as a drug reservoir for continuous MLX delivery, chitosan, with its ocular muco-adhesiveness characteristics, provides prolonged contact and release time. On the contrary, the MLX-eye drop solution was rapidly removed from the eye by tears and nasolacrimal drainage, limiting its anti-inflammatory activity.

#### 3.3.3. Pathohistological Examination

Light microscopic examination of the ocular membrane after staining with hematoxylin and eosin is shown in Figure 6. Tissue samples from the cornea of the inflamed untreated eye showed significant epithelial damage with large numbers of inflammatory cells and ulcer formation. The histopathological appearance of the corneal epithelium is considered grade 4 according to the semi-quantitative system grades. In addition, the sclera showed considerable edema and few inflammatory cells. After treatment for three days using MLX-CS-NPs eye drop dispersion, the histopathological appearance showed normal corneal epithelium with very mild inflammatory cells. In addition, the appearance of the scleral epithelium layer was normal, with no observed edema.

## 4. Conclusions

In summary, we developed a novel MLX-CS-NPs as eye drop dispersion to treat ocular inflammation. MLX was solubilized and incorporated into chitosan nanoparticles. The selected MLX-CS-NPs (F3) showed a small average particle size with a good polydispersity index, high drug entrapment, and sustained drug release. It is worth mentioning that chitosan nanotechnology enhanced the ocular contact time of MLX and improved anti-inflammatory activity with no irritation. Therefore, MLX-CS-NPs eye drop dispersion could be a promising delivery system for managing ocular inflammation with more patient compliance.

## Figures and Tables

**Figure 1 pharmaceutics-14-00893-f001:**
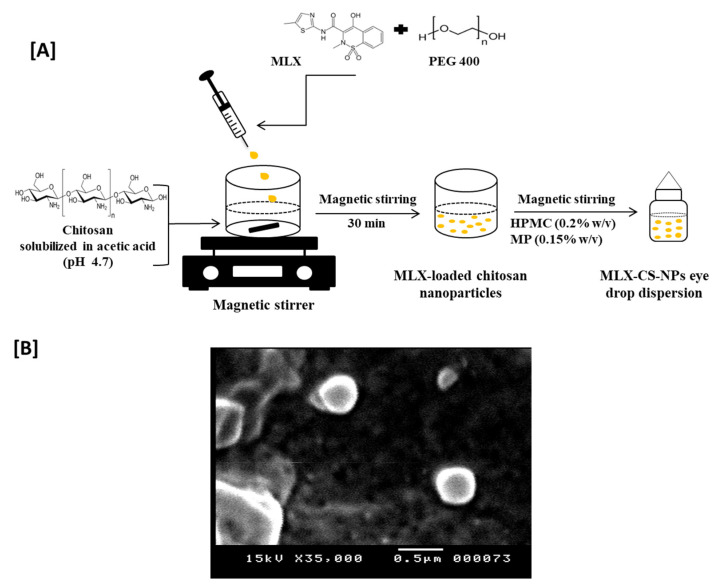
(**A**) Schematic diagram of the preparation method for MLX-CS-NPs eye drop dispersion. (**B**) Representative SEM images of the selected MLX-CS-NPs (F3). Abbreviations: MLX-CS-NPs; MLX/chitosan nanoparticles.

**Figure 2 pharmaceutics-14-00893-f002:**
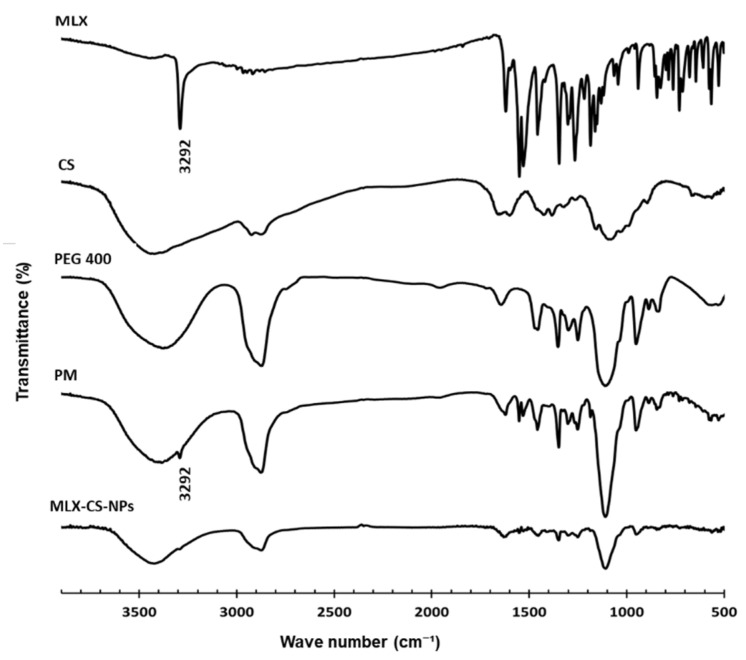
FTIR spectra of meloxicam (MLX), chitosan (CS), PEG 400, their physical mixture (PM), and MLX-CS-NPs. Abbreviations: MLX-CS-NPs; MLX/chitosan nanoparticles.

**Figure 3 pharmaceutics-14-00893-f003:**
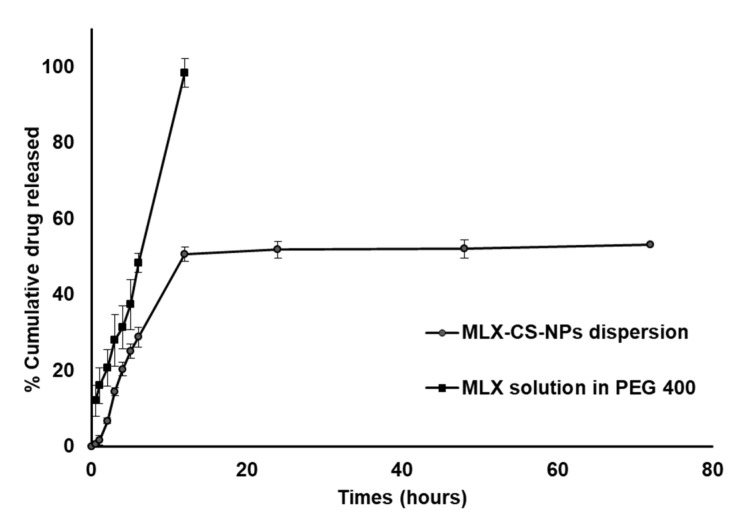
In vitro release profile of MLX-CS-NPs dispersion and MLX solution in PEG 400 using PBS (pH 7.4) as the release medium. Data are presented as the mean ± SD (*n* = 3). Abbreviations: MLX-CS-NPs; MLX/chitosan nanoparticles.

**Figure 4 pharmaceutics-14-00893-f004:**
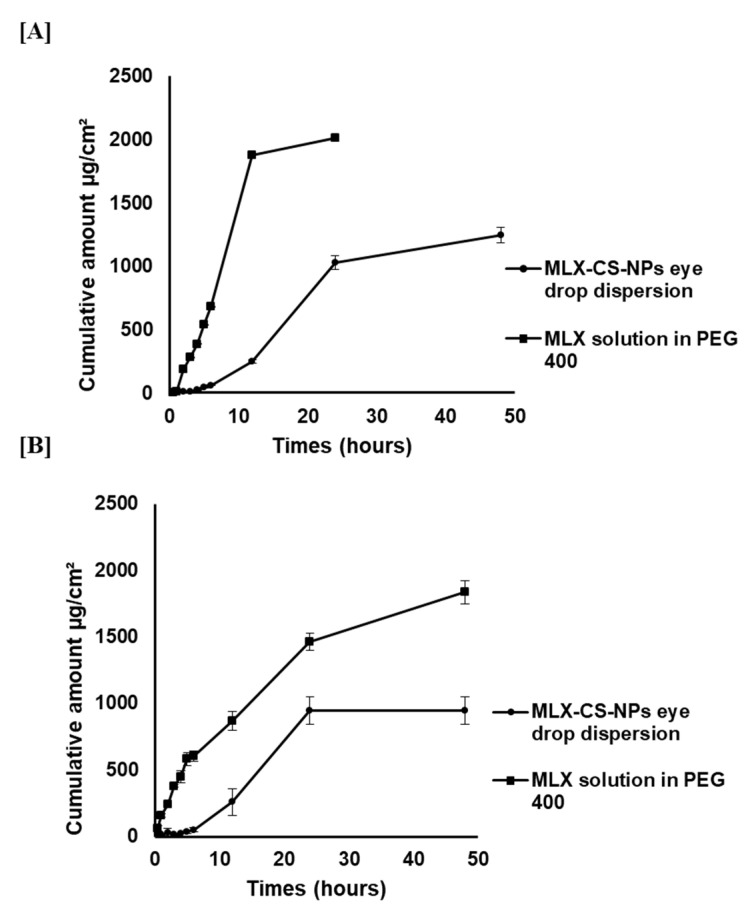
Ex vivo permeation of MLX from MLX-CS-NPs dispersion and MLX solution in PEG 400 (**A**) trans-corneal permeation and (**B**) trans-scleral permeation, data are presented as the mean ± SD (*n* = 3). Abbreviations: MLX-CS-NPs; MLX/chitosan nanoparticles.

**Figure 5 pharmaceutics-14-00893-f005:**
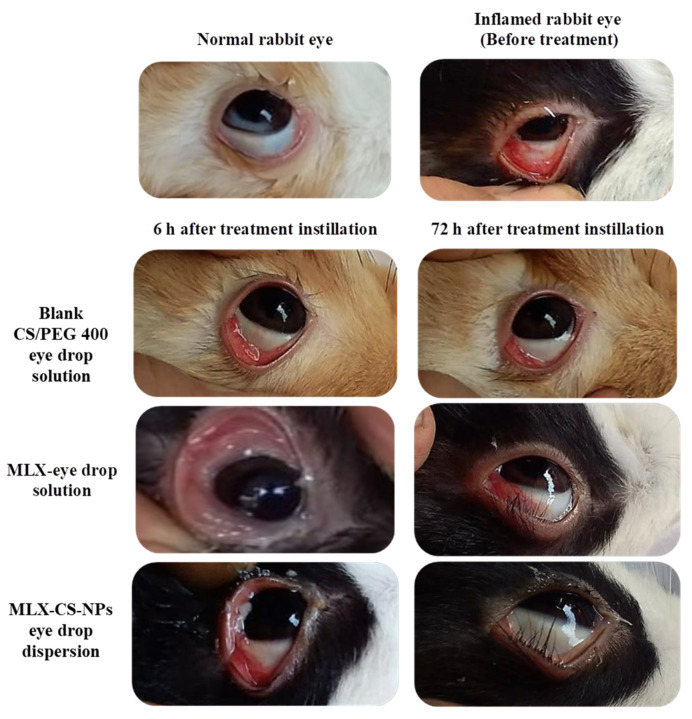
Successive images of representative eyes of rabbits before and after treatment using blank CS/PEG 400 eye drop solution, MLX-eye drop solution, and MLX-CS-NPs eye drop dispersion. Abbreviations: MLX-CS-NPs; MLX/chitosan nanoparticles.

**Figure 6 pharmaceutics-14-00893-f006:**
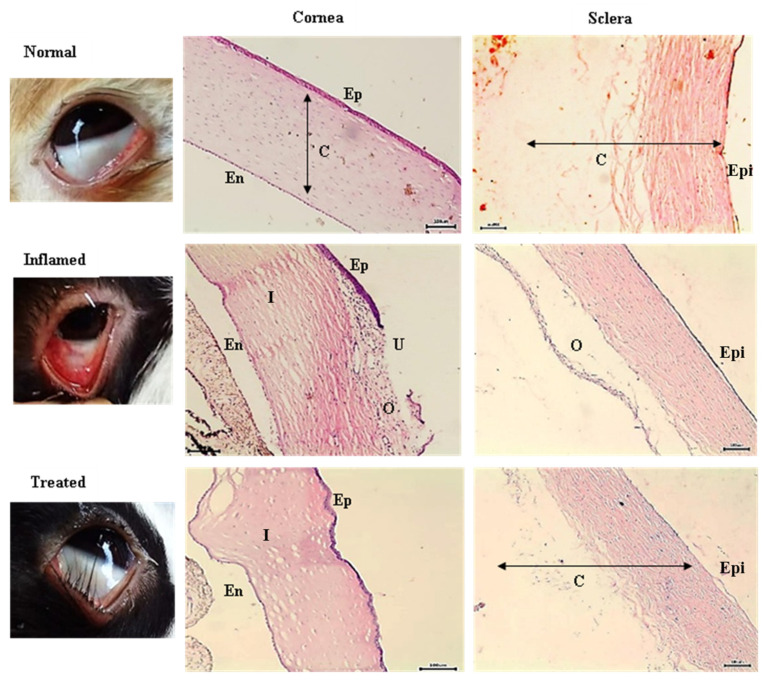
Light microscopy images of representative cornea and sclera of rabbit, before and after treatment using MLX-CS-NPs eye drop dispersion. Abbreviations; Ep, epithelium; En, endothelium; C, connective tissue; Epi, episclera; U, ulcer; O, oedema; I, inflammatory cells. Abbreviations: MLX-CS-NPs; MLX/chitosan nanoparticles.

**Table 1 pharmaceutics-14-00893-t001:** Different systems that were used in the determination of MLX solubility at 37 °C.

System Code			Polymers			
	PG (% *w*/*v*)	HPβ-CD (% *w*/*v*)	PVP (% *w*/*v*)	HPβ-CD (10% *w*/*v*): PF-127 (5% *w*/*v*)	TPP (% *w*/*v*)	PEG 400 (%)
A1	10	------	------	------	------	------
A2	20	------	------	------	------	------
A3	100	------	------	------	------	------
A4	------	1	------	------	------	------
A5	------	2.5	------	------	------	------
A6	------	5	------	------	------	------
A7	------	10	------	------	------	------
A8	------	------	------	9:1	------	------
A9	------	------	------	8:2	------	------
A10	------	------	------	7:3	------	------
A11	------	------	------	6:4	------	------
A12	------	------	------	5:5	------	------
A13	------	------	------	4:6	------	------
A14	------	------	------	3:7	------	------
A15	------	------	------	2:8	------	------
A16	------	------	------	1:9	------	------
A17	------	------	------	------	0.1	------
A18	------	------	------	------	0.25	------
A19	------	1	1	------	------	------
A20	------	2	1	------	------	------
A21	------	3	1	------	------	------
A22	------	4	1	------	------	------
A23	------	5	1	------	------	------
A24	------	6	1	------	------	------
A25	------	7	1	------	------	------
A26	------	8	1	------	------	------
A27	------	9	1	------	------	------
A28	------	10	1	------	------	------
A29	------	------	------	------	------	100

Note: The mentioned percent of polymer is the specific amount of polymer solubilized in distilled water. Abbreviations: MLX, meloxicam; PG, Propylene Glycol; HPβ-CD, Hydroxypropyl-β-cyclodextrin; PVP, Polyvinyl-pyrrolidone; PF-127, Pluronic F-127; TPP, Tripolyphosphate; PEG 400, Polyethylene glycol 400.

**Table 2 pharmaceutics-14-00893-t002:** Compositions of the prepared MLX-CS-NPs.

Formulation Number	Chitosan (% *w*/*v*)	Acetic Acid (% *v*/*v*)	MLX (mg)	0.25% *w*/*v* TPP (mL)	PEG 400 (mL)
F1	0.5	1	1	1	-
F2	0.5	1	1.5	1	-
F3	0.5	1	3.7	-	1
F4	0.25	0.5	1	1	-
F5	0.25	0.5	1.5	1	-
F6	0.25	0.5	3.7	-	1

Abbreviations: MLX-CS-NPs; MLX/chitosan nanoparticles, MLX, meloxicam; TPP, Tripolyphosphate; PEG 400, Polyethylene glycol 400.

**Table 3 pharmaceutics-14-00893-t003:** The scoring system for ocular irritation testing (OECD guidelines).

Score	Discomfort	Cornea	Conjunctiva	Discharge	Lids
0	No reaction	No changes	No changes	None	No edema
1	Blinking	Mild opacity	− Mild hyperemia − Mild edema	Mild, without wetted hair	Mild edema
2	− Enhanced blinking − Severe tearing − Vocalizations	Intense opacity	− Intense hyperemia − Intense edema − Hemorrhage	Intense, with wetted hair	Observed edema

Abbreviations: OECD, Organization for Economic Cooperation and Development.

**Table 4 pharmaceutics-14-00893-t004:** Semi-quantitative system grades of the histopathological appearance of the corneal epithelium.

Assessment	Score
Normal surface epithelium with intact microvilli and tight junctions	0
Some superficial cell sloughing and pitting with reduced microvilli	1
Denuded superficial cells with intact underlying cells	2
Partial loss of wing cells in the middle epithelial layer	3
Loss of outermost epithelial cells exposing the basal epithelial cells	4

**Table 5 pharmaceutics-14-00893-t005:** Apparent solubility of MLX in different systems.

Systems	Apparent MLX Solubility at 37 °C (mg/mL)	Comments
A1 (10% PG)	0.027	Increase MLX solubility with increasing PG concentration
A2 (20% PG)	0.05	
A3 (100% PG)	0.28	
A4 (1% HP-β-CD)	0.022	Increase MLX solubility with increasing HP-β-CD concentration
A5 (2.5% HP-β-CD)	0.048	
A6 (5% HP-β-CD)	0.095	
A7 (10% HP-β-CD)	0.18	
A8 (10% HP-β-CD:5% PF-127) (9:1)	0.15	Decrease MLX solubility with decreasing the amount of HP-β-CD
A9 (10% HP-β-CD:5% PF-127) (8:2)	0.089	
A10 (10% HP-β-CD:5% PF-127) (7:3)	0.082	
A11 (10% HP-β-CD:5% PF-127) (6:4)	0.076	
A12 (10% HP-β-CD:5% PF-127) (5:5)	0.072	
A13 (10% HP-β-CD:5% PF-127) (4:6)	0.068	
A14 (10% HP-β-CD:5% PF-127) (3:7)	0.062	
A15 (10% HP-β-CD:5% PF-127) (2:8)	0.058	
A16 (10% HP-β-CD:5% PF-127) (1:9)	0.055	
A17 (0.1%TPP)	1.3	Increase MLX solubility with increasing TPP concentration
A18 (0.25% TPP)	1.9	
A19 (1% HP-β-CD + 1% PVP)	0.026	Increase MLX solubility by increasing the percent of HP-β-CD in the presence of 1% PVP
A20 (2% HP-β-CD + 1% PVP)	0.041	
A21 (3% HP-β-CD + 1% PVP)	0.052	
A22 (4% HP-β-CD + 1% PVP)	0.072	
A23 (5% HP-β-CD + 1% PVP)	0.098	
A24 (6% HP-β-CD + 1% PVP)	0.11	
A25 (7% HP-β-CD + 1% PVP)	0.14	
A26 (8% HP-β-CD + 1% PVP)	0.16	
A27 (9% HP-β-CD + 1% PVP)	0.19	
A28 (10% HP-β-CD + 1% PVP)	0.23	
A29 (PEG 400)	3.8	Highest MLX solubility

**Table 6 pharmaceutics-14-00893-t006:** Characterization of different MLX-CS-NPs.

Formulation No.	Average Particle Size (nm)	PDI	Zeta Potential (mV)	EE (%)	pH
F1	335 ± 23	0.41± 0.0	49.2 ± 1.0	72 ± 4.5	5.6 ± 0.1
F2	597 ± 14	0.36 ± 0.1	44.4 ± 2.8	75 ± 2.0	6.2 ± 0.2
F3	195 ± 30	0.42 ± 0.0	28.2 ± 1.1	96 ± 1.5	5.6 ± 0.1
F4	266 ± 24	0.34 ± 0.1	57.0 ± 1.1	71 ± 2.0	5.5 ± 0.1
F5	493 ± 36	0.46 ± 0.0	55.9 ± 1.1	70 ± 2.5	6.3 ± 0.1
F6	242 ± 35	0.51 ± 0.0	17.3 ± 0.5	91 ± 2.0	5.5 ± 0.2

Data are presented as the mean ± SD (*n* = 3). Abbreviations: MLX-CS-NPs; MLX/chitosan nanoparticles, PDI; polydispersity index, EE; entrapment efficiency.

**Table 7 pharmaceutics-14-00893-t007:** Kinetic analysis of the in vitro release data of MLX from MLX-CS-NPs dispersion and MLX solution in PEG 400.

Formulations	Zero-Order	First-Order	Higuchi	Korsmeyer-Peppas
	R^2^	R^2^	R^2^	N
MLX-CS-NPs dispersion	0.75	0.78	0.87	0.69
MLX solution in PEG 400	0.99	0.91	0.94	0.96

Abbreviations: MLX-CS-NPs; MLX/chitosan nanoparticles.

**Table 8 pharmaceutics-14-00893-t008:** Permeation parameters of MLX through cornea and sclera of rabbit eye from MLX-CS-NPs dispersion and MLX solution in PEG 400.

Formulation	Permeation Parameters			
	Cornea		Sclera	
	J (µg·cm^−2^·h^−1^)	P (cm·h^−1^)	J (µg·cm^−2^·h^−1^)	P (cm·h^−1^)
MLX-CS-NPs dispersion	29.9	0.0199	23.7	0.0158
MLX solution in PEG 400	95.1	0.0634	36.8	0.0246

Abbreviations: MLX-CS-NPs; MLX/chitosan nanoparticles, J; steady-state flux, P; permeability coefficient.

**Table 9 pharmaceutics-14-00893-t009:** Clinical examination scoring for the inflamed eyes after applying MLX-CS-NPs eye drop dispersion and MLX-eye drop solution.

Experiment Groups	The Average Score of Eye Inflammation		
	Before the Start of Treatment	After One Day of Treatment	After Three Days of Treatment
Group I: Blank CS/PEG 400 eye drop solution	8 ± 0.3	6.3 ± 0.7	2.7 ± 0.5
Group II: MLX-eye drop solution	10 ± 0.8	7.3 ± 0.8	5.0 ± 0.7
Group III: MLX-CS-NPs eye drop dispersion.	10 ± 1.0	4.0 ± 0.5	0.3 ± 0.2

Data are presented as the mean ± SD (*n* = 3). Abbreviations: MLX-CS-NPs; MLX/chitosan nanoparticles.

## Data Availability

Not applicable.

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
