# Peer review of "Chitosan Nanoparticles for Meloxicam Ocular Delivery: Development, In Vitro Characterization, and In Vivo Evaluation in a Rabbit Eye Model"

_pharmaceutics, 2022, doi:10.3390/pharmaceutics14050893_

Round 1

Reviewer 1 Report

The manuscript titled Chitosan nanoparticles for meloxicam ocular delivery by Hebatallah is a good study and authors have performed sufficient experiments to provide the proof of concept. Technically the manuscript is good, however, the English language is not appropriate. Authors should go for English Language Editing from a person expert in pharmaceutics.

I am expressing some concerns that authors should take care of.

  1. For the entire manuscript, the English language is not proper. Giving you a few examples
    1. Conventional eye drops are therefore the most suitable route for management: ‘’therefore’’ word is not appropriate, as it has no connection to previous line
    2. To overcome this, frequent application will be needed- ‘’will be ‘’ is not correct
    3. To the best of our knowledge, this first study that developed- incomplete sentence
    4. using 0.45 μm filter membrane- ‘’through 0.45 μm membrane filter’’
  2. Fig 4 A and B- Error bars indicate SD or SEM? Also please specify n=?
  3. For permeation data analysis and calculation of various parameters, please cite reference. E.g. https://pubmed.ncbi.nlm.nih.gov/23159662/
  4. Section 2.3- Air drying has not affected the particle characteristics? Authors should have tried lyophilization or give data that dried particles had retained initial characteristics.
  5. Being ophthalmic product, isotonicity should be measured. Otherwise, osmotic pressure may affect lachrymal flow.

Author Response

The authors thank the reviewer for his/her valuable comments, and all will be considered point by point as follow:

  1. Fig 4 A and B- Error bars indicate SD or SEM? Also please specify n=?

Thank you for this remark. The error bars indicate SD with n=3 (we added this to the manuscript)

  1. For permeation data analysis and calculation of various parameters, please cite reference. E.g.https://pubmed.ncbi.nlm.nih.gov/23159662/

Thank you for the information. (We added that reference to the manuscript)

  1. Section 2.3- Air drying has not affected the particle characteristics? Authors should have tried lyophilization or give data that dried particles had retained initial characteristics.

Thank you for the comment. Nanoparticles were air-dried at room temperature (we added this to the manuscript)

  1. Being ophthalmic product, isotonicity should be measured. Otherwise, osmotic pressure may affect lachrymal flow

Thank you for the comment. We used isotonic phosphate buffered saline to prepare all eye drops.   

Reviewer 2 Report

The manuscript of Hebatallah B. Mohamed et al. entitled "Chitosan nanoparticles for meloxicam ocular delivery: development, in vitro characterization, and in vivo evaluation in a rabbit eye model” presents an ocular therapy based on the encapsulation of meloxicam drug in chitosan nanoparticles. It is a well written manuscript, the application of meloxicam-based chitosan nanoparticles is novel for ocular inflammation, but there are already several publications based of these nanoparticles (Journal of Magnetism and Magnetic Materials 20221, 523, 167571 and Kola Venu et al., Saudi J. Med. Pharm. Sci., 2018, 4, 2, 270). Although some interesting data have been presented, several issues need to be addressed and added before this manuscript could be considered for publication. I would recommend a resubmission with major revision based on the following general comments:

Line 60. Please could you add the type of PG used in the experiments and which is the different between PG and PEG400?

Line 80. It is stated that the authors do not fully understand the mechanism of formation of chitosan nanoparticles. Protonated amine (NH3+) from chitosan are complexed by OH from TPP, MLX and PEG, if there are unprotonated in acid conditions, thing I doubt for the PEG. Regarding the synthesis, there is no sense in applying 30’ magnetic stirring and then 10 min of ultrasounds probe. The key point in the nanoparticle formation is the moment when counterion is added, since the homogenization method used in this step greatly affects the size and homogeneity of the nanoparticles. To improve the homogeneity of the samples, the authors must first apply ultrasound and then allow the chitosan to cure by magnetic stirring.

Please could you also provide the synthesis, particle size, zeta potential, and SEM microscopy of your control chitosan nanoparticles without MLX formed only by chitosan and PEG400.

Line 142. In the vitro release, the ratio between the volume of release medium (50 mL) and the sample aliquots (5 mL) sample is very small, and in the course of the experiment practically 45 mL have been extracted. The authors do not believe that this can add a large error to the results obtained.

Line 253.  According to literature MLX solubility in PEG400 is around 7 mg/mL. Why the value of authors is so low in comparison? I would recommend to the authors to make a study of the effect of pH over MLX molecule, since depending on the pH, MLX can be in an anion zwitterion and cation form, and this is very important to know the form of MLX for chitosan nanoparticle formation.

Line 261. Please correct the interactions between the chitosan and the other molecules involved in the nanoparticle formation. First, MLX is solubilized by PEG400 or TPP. Next, there is an ionic complexation between negatively charged phosphate groups of TPP and MLX with positively charged amino groups of chitosan. In the case of MLX/PEG400@chitosan nanoparticles, there is only interaction between MLX and chitosan by ionic complexation for forming the nanoparticles. The role of PEG400 is to keep MLX solubilized. Please could you provide any data that chitosan and PEG400 alone can form nanoparticles?

Line 265. Please can the authors add the composition of samples represented in Figure 1. On the other hand, according to the SEM images, the authors cannot confirm that they have spherical uniform, and smooth surface nanoparticles. The represented prototypes look like a physical mixture of the reactives. This is a very important point; the authors must provide a good image to confirm the formation of the nanoparticles.

Line 279. According to literature, a moderate stability in nanoparticles is achieved at zeta potential >+30 mV or <-30mV (https://doi.org/10.1016/B978-0-08-100557-6.00003-1). For the prototypes based on PEG400, the authors are in the limit.  

Line 285. According to the PDI values, samples are not uniform. Values of 0.2 and below are most commonly deemed acceptable in practice for polymer-based nanoparticle material.  A value of PDI of 0.42 is not acceptable because this material will have stability problems.

Line 296. I would recommend to the author to revise this apart. For example, 3292 cm-1 is not associated to the hydroxyl group of MLX, but it is related to the MLX secondary amine stretching. Please have a look to this reference (Journal of Magnetism and Magnetic Materials 20221, 523, 167571) where it is well explained the FT-IR spectra of MLX and chitosan.

Line 384. Do you have any explanation why blank chitosan NPs present an eye inflammation score after 3 days of treatment lower than the MLX drop solution?

Author Response

The authors thank the reviewer for his/her valuable comments and all will be considered point by point as follow:

  • Line 60…….. Please could you add the type of PG used in the experiments and which is the different between PG and PEG400?

Thank you.

  • PG (0.995 mass fraction purity) (It was added to the manuscript)
  • Propylene glycol (propane-1,2-diol) containing two alcohol groups and Polyethylene glycol is a polyethylene oxide, both polymers have been used to solubilize hydrophobic drugs.

  • Line 80……. It is stated that the authors do not fully understand the mechanism of formation of chitosan nanoparticles. Protonated amine (NH3+) from chitosan are complexed by OH from TPP, MLX and PEG, if there are unprotonated in acid conditions, thing I doubt for the PEG. Regarding the synthesis, there is no sense in applying 30’ magnetic stirring and then 10 min of ultrasounds probe. The key point in the nanoparticle formation is the moment when counterion is added, since the homogenization method used in this step greatly affects the size and homogeneity of the nanoparticles. To improve the homogeneity of the samples, the authors must first apply ultrasound and then allow the chitosan to cure by magnetic stirring.

Please could you also provide the synthesis, particle size, zeta potential, and SEM microscopy of your control chitosan nanoparticles without MLX formed only by chitosan and PEG400.

Thank you for this comment.

  • There is no nanoparticle formed by chitosan and PEG400. In our study MIX act as a counterion due to it is an anionic in nature. Please, see this reference … (DOI:5562/cca3346).
  • When MLX/PEG400 solution was added into the CS/acetic acid solution, complexation occurred between MLX and chitosan through hydrogen bonding and Van der Waals interaction that was led to formation of MLX-CS-NPs.

  • Line 142…… In the vitro release, the ratio between the volume of release medium (50 mL) and the sample aliquots (5 mL) sample is very small, and in the course of the experiment practically 45 mL have been extracted. The authors do not believe that this can add a large error to the results obtained

Thank you for this comment.

  • Although the reported solubility of meloxicam in water is very low and it is considered practically insoluble, this solubility increases and reached to 2.236 mg/ml in 0.5 M phosphate buffered saline pH 7.4, Please see this reference….

 https://patents.google.com/patent/WO2014161131A1/en.

  • We only used total amount of MLX in the nanoparticles equal to 3.5 mg and the solubility in PBS pH 7.4 equal to 2.2 mg/ml; the used total amount of PBS is 50 mL. We believe that this volume of buffer could maintain the sink condition with 5 ml fresh PBS replacement each time.
  • We followed the same previously reported in-vitro release experiment. Please see the reference ……… DOI:9734/BJPR/2016/26985

  • Line 253…..  According to literature MLX solubility in PEG400 is around 7 mg/mL. Why the value of authors is so low in comparison? I would recommend to the authors to make a study of the effect of pH over MLX molecule, since depending on the pH, MLX can be in an anion zwitterion and cation form, and this is very important to know the form of MLX for chitosan nanoparticle formation.

Thank you for this comment.

  • This may be due to differences in drug polymorphic states, equilibration times, and other experimental conditions.
  • The solubility of meloxicam was previously reported as equal to 3.763mg/ml. Please, see this reference….. doi: 1208/pt040333.

  • Line 261…… Please correct the interactions between the chitosan and the other molecules involved in the nanoparticle formation. First, MLX is solubilized by PEG400 or TPP. Next, there is an ionic complexation between negatively charged phosphate groups of TPP and MLX with positively charged amino groups of chitosan. In the case of MLX/PEG400@chitosan nanoparticles, there is only interaction between MLX and chitosan by ionic complexation for forming the nanoparticles. The role of PEG400 is to keep MLX solubilized. Please could you provide any data that chitosan and PEG400 alone can form nanoparticles?

  • Thank you for this information. There is no nanoparticle observed from CS/PEG400. There is only interaction between MLX and chitosan by ionic complexation for forming the nanoparticles. As MLX itself act as the anionic cross-linker and this process can be named ‘drug-induced gelation’ like other previously reported study …………(we added this part to the manuscript)
  • Please see the references….
  • DOI: 1111/jphp.12076
  • DOI: 10.5562/cca3346
  • We replaced (chitosan blank nanoparticles) expression to blank chitosan/PEG400 solution (CS/PEG400) in the manuscript.

  • Line 265…... Please can the authors add the composition of samples represented in Figure 1. On the other hand, according to the SEM images, the authors cannot confirm that they have spherical uniform, and smooth surface nanoparticles. The represented prototypes look like a physical mixture of the reactives. This is a very important point; the authors must provide a good image to confirm the formation of the nanoparticles.

Thank you for this comment. We apologize for this unclear presentation. The two images in figure 1 are representative images for MLX-CS-NPs and there is no SEM image for physical mixture reported here……(We added a more clear SEM images to the manuscript)

  • Line 279…… According to literature, a moderate stability in nanoparticles is achieved at zeta potential >+30 mV or <-30mV (https://doi.org/10.1016/B978-0-08-100557-6.00003-1). For the prototypes based on PEG400, the authors are in the limit.

Thank you for this comment. We added 0.2% w/v hydroxypropyl methylcellulose (HPMC) as viscolizer to the eye drop dispersion to prevent agglomeration.

  • Line 285….. According to the PDI values, samples are not uniform. Values of 0.2 and below are most commonly deemed acceptable in practice for polymer-based nanoparticle material.  A value of PDI of 0.42 is not acceptable because this material will have stability problems.

Thank you for the remark. In our study, the PDI values were within the acceptable range. Previous studies reported that PDI values greater than 0.7 are indicative for a very broad size distribution of the sample. Please see references…

  • https://doi.org/10.1016/j.carbpol.2020.116878
  • doi:10.1088/1757-899X/311/1/012024

  • Line 296….. I would recommend to the author to revise this apart. For example, 3292 cm-1 is not associated to the hydroxyl group of MLX, but it is related to the MLX secondary amine stretching. Please have a look to this reference (Journal of Magnetism and Magnetic Materials 20221, 523, 167571) where it is well explained the FT-IR spectra of MLX and chitosan.

  • Thank you for this information (It was corrected in the manuscript).

  • Line 384….. Do you have any explanation why blank chitosan NPs present an eye inflammation score after 3 days of treatment lower than the MLX drop solution?

Thank you for this comment. Chitosan nanoparticles has some anti-inflammatory effect that’s why we used it as a delivery system for MLX. In our research we highlighted the importance of the presence of chitosan with MLX for an enhanced anti-inflammatory activity. That was proved experimentally in this in vivo study.

Round 2

Reviewer 2 Report

I thank the authors for the additional explanations and clarifications. I do believe the manuscript presents an interesting study, but I would not recommend publishing the results in Pharmaceutics, or any other journal, before thoroughly validating the morphology results of chitosan nanoparticles, especially the SEM images shown in Figure 1. In the first revision, the authors have present the same image and an extension of itself, and with this image we cannot confirm that nanoparticles are well-formed . I would therefore recommend a major revision of the manuscript.

Author Response

Please see to attachment.
